# Serum RNAs can predict lung cancer up to 10 years prior to diagnosis

Sinan U Umu[1]*, Hilde Langseth[1,2], Verena Zuber[2], Åslaug Helland[3,4,5], Robert Lyle[6,7], Trine B Rounge[1,8]*

[1]Department of Research, Cancer Registry of Norway, Oslo, Norway; [2]Department of Epidemiology and Biostatistics, Imperial College London, London, United Kingdom; [3]Department of Oncology, Oslo University Hospital, Oslo, Norway; [4]Institute for Cancer Research, Oslo University Hospital, Oslo, Norway; [5]Institute of Clinical Medicine, University of Oslo, Oslo, Norway; [6]Department of Medical Genetics, Oslo University Hospital and University of Oslo, Oslo, Norway; [7]Centre for Fertility and Health, Norwegian Institute of Public Health, Oslo, Norway; [8]Department of Informatics, University of Oslo, Oslo, Norway

*For correspondence:
sinan.ugur.umu@kreftregisteret.
no (SUU);
trine.rounge@kreftregisteret.
no (TBR)

**Competing interest:** The authors declare that no competing interests exist.

**Abstract** Lung cancer (LC) prognosis is closely linked to the stage of disease when diagnosed. We investigated the biomarker potential of serum RNAs for the early detection of LC in smokers at different prediagnostic time intervals and histological subtypes. In total, 1061 samples from 925 individuals were analyzed. RNA sequencing with an average of 18 million reads per sample was performed. We generated machine learning models using normalized serum RNA levels and found that smokers later diagnosed with LC in 10 years can be robustly separated from healthy controls regardless of histology with an average area under the ROC curve (AUC) of 0.76 (95% CI, 0.68–0.83). Furthermore, the strongest models that took both time to diagnosis and histology into account successfully predicted non-small cell LC (NSCLC) between 6 and 8 years, with an AUC of 0.82 (95% CI, 0.76–0.88), and SCLC between 2 and 5 years, with an AUC of 0.89 (95% CI, 0.77–1.0), before diagnosis. The most important separators were microRNAs, miscellaneous RNAs, isomiRs, and tRNA-derived fragments. We have shown that LC can be detected years before diagnosis and manifestation of disease symptoms independently of histological subtype. However, the highest AUCs were achieved for specific subtypes and time intervals before diagnosis. The collection of models may therefore also predict the severity of cancer development and its histology. Our study demonstrates that serum RNAs can be promising prediagnostic biomarkers in an LC screening setting, from early detection to risk assessment.

## Editor's evaluation

This work has generated valuable data demonstrating the potential utility of serum RNA for lung cancer detection.

## Introduction

Lung cancer (LC) continues to be the leading cause of cancer-related deaths despite declining smoking prevalence (*Bray et al., 2018*; *Wild et al., 2020*). Non-small-cell (NSCLC) and small-cell (SCLC) are the two major subtypes of LC. The symptoms generally occur at a late stage and the prognosis is poor. Stage at diagnosis typically determines patient survival (*Aberle et al., 2011*; *Bach et al., 2012*; *Brustugun et al., 2018*). Screening with low-dose computed tomography (LDCT) can be effective for early detection (*Bach et al., 2012*; *Peled and Ilouze, 2015*) and reduce LC mortality up to 20% in

high-risk groups (*de Koning et al., 2020*; *Hanash et al., 2018*; *Seijo et al., 2019*). However, LDCT has limitations such as high false-positive rates, risk of overdiagnosis, and high costs (*Gopal et al., 2010*; *Peled and Ilouze, 2015*). Annual CT scans also cause harmful radiation exposure (*Bach et al., 2012*; *Hanash et al., 2018*). Robust biomarkers can help stratify high-risk groups and increase accuracy in patient inclusion criteria for LDCT-based screening programs (*Hanash et al., 2018*).

Liquid biopsies quantifying molecular biomarkers in circulation, such as tumor-derived DNAs, proteins, and RNAs, can be used to detect cancer (*Hanash et al., 2018*; *Ko et al., 2018*; *Sandfeld-Paulsen et al., 2016*). MicroRNAs (miRNA), a class of ~21 nucleotide long short RNAs, have been widely investigated for their biomarker potential (*Fehlmann et al., 2020*; *Keller and Meese, 2016*; *Pichler and Calin, 2015*; *Tian et al., 2019*). They can be found both in serum (*Keller and Meese, 2016*; *Murillo et al., 2019*; *Umu et al., 2018*) and in plasma (*Freedman et al., 2016*; *Keller and Meese, 2016*; *Murillo et al., 2019*) as cell-free circulating RNAs, which may originate from dying cells or be actively secreted (*Zaporozhchenko et al., 2018*). Some of them are bounded by proteins or confined in layered exosomes which can protect them from degradation (*Fritz et al., 2016*). MiRNAs can function as tumor suppressors or oncomiRs and regulate tumor traits such as cell growth, angiogenesis, immune evasion, and metastasis (*Pichler and Calin, 2015*; *Svoronos et al., 2016*). The search for RNA biomarkers is not limited to miRNAs. Aberrant expression of other RNA classes, such as protein coding mRNAs, tRNAs, piwi-interacting RNAs (piRNAs), and long-noncoding RNAs (lncRNAs), has been associated with cancer (*Kim et al., 2017*; *Slack and Chinnaiyan, 2019*). Despite the immense potential of cell-free RNAs, the promise of non-invasive RNA biomarkers of cancer has not yet been fulfilled.

One explanation of the lack of circulating RNAs used in clinical settings is our limited understanding of the prediagnostic dynamics of cell-free RNAs, since studies are usually based on samples at or after diagnosis. Carcinogenesis is a multistep process that turns cell functions from normal to malignant (*Hanahan and Weinberg, 2000*). It can cause temporal changes in RNA levels linked to

**Table 1.** Clinical and histological characteristics of samples used in modeling.

| | | Stage | | | |
|---|---|---|---|---|---|
| | Early (localized) | Locally Advanced (regional) | Advanced (distant) | Unknown | Controls |
| **Histology** | | | | | |
| NSCLC | 84 | 99 | 167 | 11 | - |
| SCLC | 9 | 35 | 76 | 4 | - |
| Others | 10 | 5 | 31 | 4 | - |
| **Sex** | | | | | |
| Male | 78 | 104 | 178 | 12 | 185 |
| Female | 25 | 35 | 96 | 7 | 78 |
| **Age at donation**, years | | | | | |
| Mean (SD) | 54.3 (7.33) | 54.9 (9.08) | 53.5 (8.25) | 51.8 (6.53) | 49.9 (10.9) |
| **Age at diagnosis**, years | | | | | |
| Mean (SD) | 59.8 (7.67) | 60.6 (8.89) | 59.4 (8.31) | 58.6 (6.05) | - |
| **Prediagnostic sampling time**, years | | | | | |
| Mean (SD) | 5.52 (2.81) | 5.63 (2.79) | 5.91 (2.66) | 6.75 (2.18) | - |
| **Total samples** | 103 | 139 | 274 | 19 | 263 |
| **Individuals** | 79 | 102 | 189 | 16 | 263 |
| *Total individuals* | | | 645 (smokers*) | | |

*See supplementary document for non-smokers (*Supplementary file 1*).

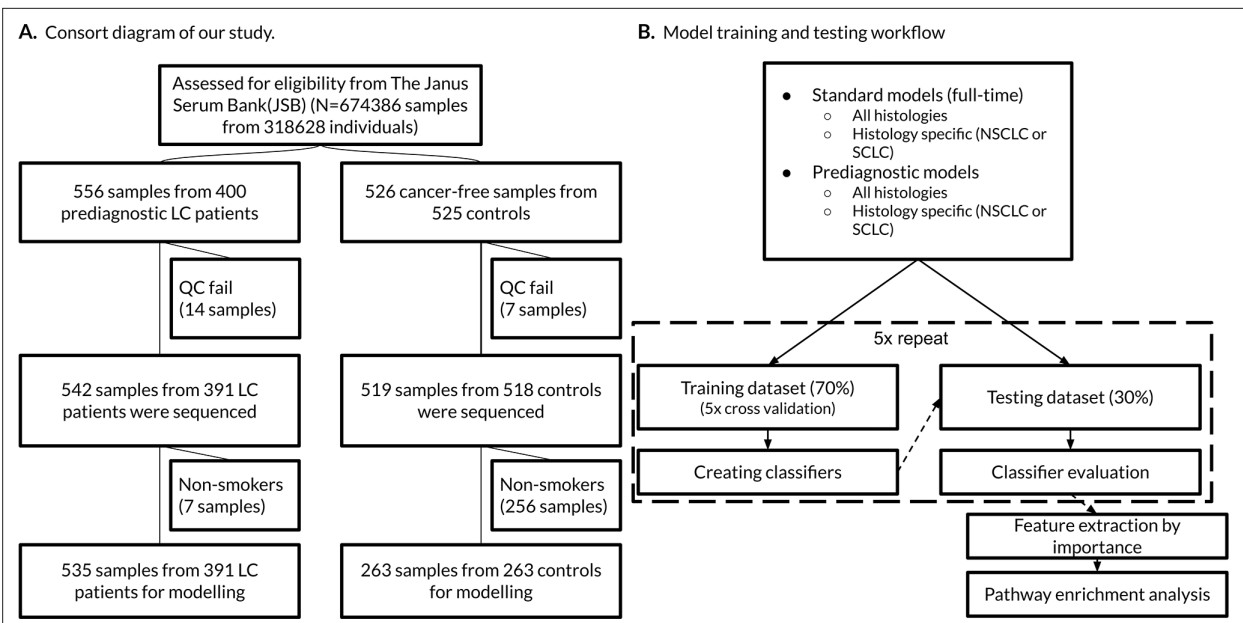

**Figure 1.** Consort diagram of the study and our model training and testing workflow. (**A**) The sample selection is summarized by the flow chart. Non-smokers were excluded from model building. (**B**) We randomly created five different training and testing datasets for each group (e.g. standard, histology-specific, or prediagnostic models).

cellular processes driven by the hallmarks of cancer (*Gutschner and Diederichs, 2012*; *Hanahan and Weinberg, 2000*). We have shown that prediagnostic RNA levels in serum are highly dynamic in LC patients, which may signal early carcinogenesis (*Umu et al., 2020*). A similar result was observed in breast cancer (*Lund et al., 2016*) and testicular cancer patients (*Burton et al., 2020*). A lack of reproducibility among studies is also a problem, caused by technical and biological factors such as storage time, sampling procedure, age, sex, smoking history, etc. (*Rounge et al., 2018*). It is therefore important to control for these factors.

In the present study, our objective was to identify serum RNA-based biomarkers for early diagnosis of LC using prediagnostic samples. We identified the optimal machine learning (ML) algorithm for RNA biomarker modeling. Optimization of prediction models was done with an ML workflow, including cross-validation and testing, which was repeated five times to increase the generalizability of our results. We also investigated the biological relevance of the best RNA separators in the context of cancer biomarkers.

## Results

### Patient characteristics and RNA-seq profiles

In this study, we selected 400 patients with prediagnostic serum samples including multiple samples from the same patients. We also included 525 individuals as controls. After excluding failed or low input samples, we obtained RNA-seq data from 1061 serum samples. However, samples from individuals without any smoking history (i.e. never smokers) or missing information were excluded from initial analyses. This resulted in 535 cases and 263 control samples from 645 current or former smokers for modeling and testing (*Table 1* and *Figure 1A*). Non-smokers consist of 7 cases and 256 control samples from 260 individuals (*Supplementary file 1*). We used non-smokers in a leave-out set only to test our final models and to calculate relative risk (RR).

After filtering out low-count transcripts, 3306 RNAs were selected as candidate features and used in the models: 202 miRNAs, 1137 isomiRs, 89 miscellaneous RNAs (miscRNAs), 380 piRNAs, 119 small nucleolar RNAs, 530 tRFs, 790 mRNAs, and 59 lncRNAs.

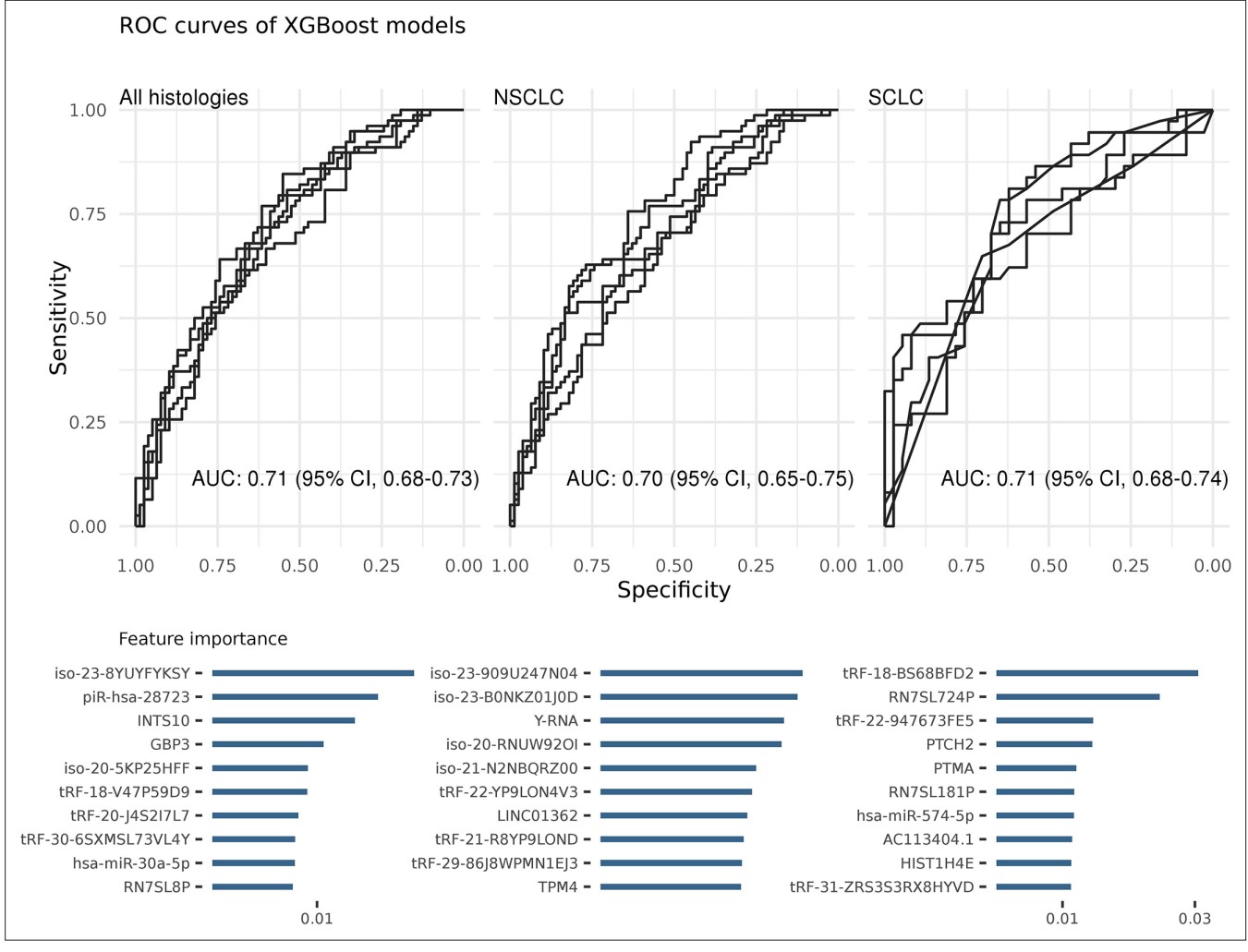

**Figure 2.** Each ROC curve is based on the prediction results of a randomly created testing dataset (in total five). Area under the ROC curve (AUC) values show the average of these predictions. The most important features of the classifiers were sorted on their average feature importance and are shown in the lower panels. A detailed list of biomarkers with their feature importance is available in supplementary (*Supplementary file 2*). We did not perform any feature selection while training these models (see also *Figure 2—source data 1*).

The online version of this article includes the following source data and figure supplement(s) for figure 2:

**Source data 1.** Source data of XGBoost ROC plots for *Figure 2*.

**Figure supplement 1.** Each boxplot shows performances of an algorithm measured by area under the ROC curves (AUCs).

**Figure supplement 1—source data 1.** Source data of boxplots for *Figure 2—figure supplement 1*.

**Figure supplement 2.** ROC curves of various types of models with/without serial samples.

**Figure supplement 2—source data 1.** Source data of ROC plots without multiple samples from same individuals (*Figure 2—figure supplement 2*).

## ML algorithms can differentiate between prediagnostic cases and controls regardless of prediagnostic time

We first evaluated the classification performance of the ML algorithms in terms of average AUCs on test datasets, created by five random repeats as explained in Materials and methods.

All samples were included in algorithm evaluation regardless of their stage at diagnosis and prediagnostic time which were regarded as full-time standard models (*Figure 2—figure supplement 1*). The average AUC of all algorithms was 0.67 (95% CI, 0.66–0.69) for all histologies, 0.67 (95% CI, 0.65–0.69) for NSCLC and 0.64 (95% CI, 0.62–0.66) for SCLC on the test datasets. The XGBoost algorithm produced a higher AUC than the average, 0.71 (95% CI, 0.68–0.73). The XGBoost models also

performed better when the samples were stratified by histologies: NSCLC, 0.70 (95% CI, 0.65–0.75) and SCLC, 0.71 (95% CI, 0.68–0.74) (*Figure 2*).

Although the models of all algorithms had comparable performances in terms of average AUCs, they differ in terms of total number of non-zero features (i.e. different model complexity). For example, random forest (RF) selected more than 3000 non-zero features while the lasso model selected fewer than 25 features. However, the profiles of the top features, ranked in terms of feature importance, usually consisted of similar RNAs (e.g. miRNAs or tRFs).

Since XGBoost produced the most predictive full-time models, we used it for the remaining analyses. We also investigated the best predictors of the XGBoost models and ranked them based on their importance (*Supplementary file 2*). The top three best features were an isomiR of hsa-miR-486–5p (iso-23-8YUYFYKSY), piR-hsa-28723, and *INTS10* for all histologies; Y-RNA, piR-hsa-28723, and *GPB3* for NSCLC; and tRF-BS68BFD2, RN7SL724P, and tRF-947673FE5 for SCLC. An in-depth investigation of selected features by other algorithms also showed common RNAs. For example, Y-RNA and iso-23-8YUYFYKSY isomiR were among the top predictors of the RF, elastic-net, the SGL, and the lasso models for NSCLC; tRF-BS68BFD2 for SCLC. We also performed KEGG pathway enrichment analysis based on the common miRNA, mRNA, and isomiR features. The results showed that many cancer-related pathways were significantly (p < 0.01) enriched such as MAPK signaling, mTOR signaling, and AMPK signaling.

We evaluated the classification performance of the XGBoost algorithm by selecting one sample per patient rather than using all samples from the same individuals. Our results showed comparable performance in terms of AUCs for all models (*Figure 2—figure supplement 2*). The SCLC models performed slightly worse than the others. This discrepancy can be explained by the relatively small sample size of this group. Therefore, we decided to use all samples from the same individuals.

## MiscRNA- and miRNA-only models are more accurate than the others

We produced XGBoost models that included only a single RNA class (e.g. miRNA, isomiR, etc.) to further investigate important features/classes. This method showed that miscRNA-only and miRNA-only models achieved better classification performance than the other RNA classes regardless of histology and stage at diagnosis (*Table 2*). The best separators of these models included hsa-miR-99a-5p, hsa-miR-1908–5p, hsa-miR-3925–5p, and Y-RNA-related transcripts

**Table 2.** Averages of area under the ROC curves (AUCs), accuracies (acc), sensitivities (sn), and specificities (sp) of the XGBoost algorithm models on test datasets when prediagnostic time was not included.

| Features included: | **All (including others)** | | | **NSCLC** | | | **SCLC** | | |
|---|---|---|---|---|---|---|---|---|---|
| | AUC | Av. # of features* | Av. % of acc/sn/sp | AUC | Av. # of features | Av. % acc/sn/sp | AUC | Av. # of features | Av. % acc/sn/sp |
| All RNAs | 0.71 (95% CI, 0.68–0.73) | 301 | 69/73/62 | 0.70 (95% CI, 0.65–0.75) | 373 | 67/70/64 | 0.71 (95% CI, 0.68–0.74) | 213 | 70/69/71 |
| Lasso-selected features | 0.78 (95% CI, 0.74–0.82) | 149 | 73/75/71 | 0.78 (95% CI, 0.75–0.82) | 56 | 73/73/72 | 0.74 (95% CI, 0.69–0.80) | 58 | 72/61/83 |
| Univariate significant features | 0.70 (95% CI, 0.66–0.73) | 76 | 67/75/58 | 0.69 (95% CI, 0.64–0.73) | 51 | 67/71/64 | 0.70 (95% CI, 0.65–0.76) | 11 | 68/69/68 |
| miRNA only | 0.72 (95% CI, 0.68–0.76) | 168 | 69/76/61 | 0.73 (95% CI, 0.70–0.75) | 199 | 69/74/64 | 0.65 (95% CI, 0.62–0.69) | 20 | 67/74/60 |
| isomiR only | 0.70 (95% CI, 0.65–0.74) | 204 | 67/68/67 | 0.73 (95% CI, 0.69–0.77) | 215 | 71/75/66 | 0.65 (95% CI, 0.60–0.70) | 108 | 66/65/67 |
| tRF only | 0.69 (95% CI, 0.65–0.73) | 314 | 65/77/53 | 0.67 (95% CI, 0.65–0.69) | 314 | 66/64/67 | 0.68 (95% CI, 0.65–0.71) | 23 | 66/69/63 |
| MiscRNA only | 0.72 (95% CI, 0.69–0.74) | 83 | 69/73/65 | 0.68 (95% CI, 0.63–0.74) | 87 | 66/73/59 | 0.69 (95% CI, 0.64–0.75) | 76 | 70/78/61 |

*Average number of non-zero features selected by the models. *Note*: Detailed information on all selected features are in **Supplementary file 2**.

(i.e. RNY1P5 and RNY4P30). When we took histology into account, miRNAs and isomiRs for NSCLC and miscRNAs for SCLC produced better models (*Table 2*). The most important features of histology-dependent models included hsa-miR-629–5p, hsa-miR-99a-5p, hsa-miR-486–5p isomiR (iso-23-8YUYFYKSY), hsa-miR-151a-3p isomiR (iso-22-B0NKZK1JN) for NSCLC; 7SL RNA-related transcripts and vault-RNA for SCLC (*Supplementary file 2*).

## Feature selection improves model performance and reduces model complexity

Single RNA class models also implied that feature selection can further improve model performances. Thus, we tested two feature selection methods. The results showed that lasso feature selection improved AUC values and reduced complexity (*Table 2*). The most important features of lasso-selected models included hsa-miR-423–5p isomiR (iso-20-5KP25HFF), *GBP3*, and piR-hsa-28723 for all histologies; Y-RNA, hsa-miR-423–5p isomiR (iso-20-5KP25HFF), and *LINC*01362 for NSCLC; *HIST1H4E*, *PTCH2,* and tRF-R29P4P9L5HJVE for SCLC (*Supplementary file 2*). Moreover, univariate significant feature selection greatly reduced model complexity with an acceptable performance (*Table 2*). For example, SCLC models only included 11 RNAs. The most important features were *GBP3*, LINC01362, and hsa-miR-30a-5p for all histologies; LINC01362, *GBP3,* and tRF-9MV47P596V for NSCLC; piR-hsa-7001 and tRF-7343*R* × 6NMH3 for SCLC (*Supplementary file 2*).

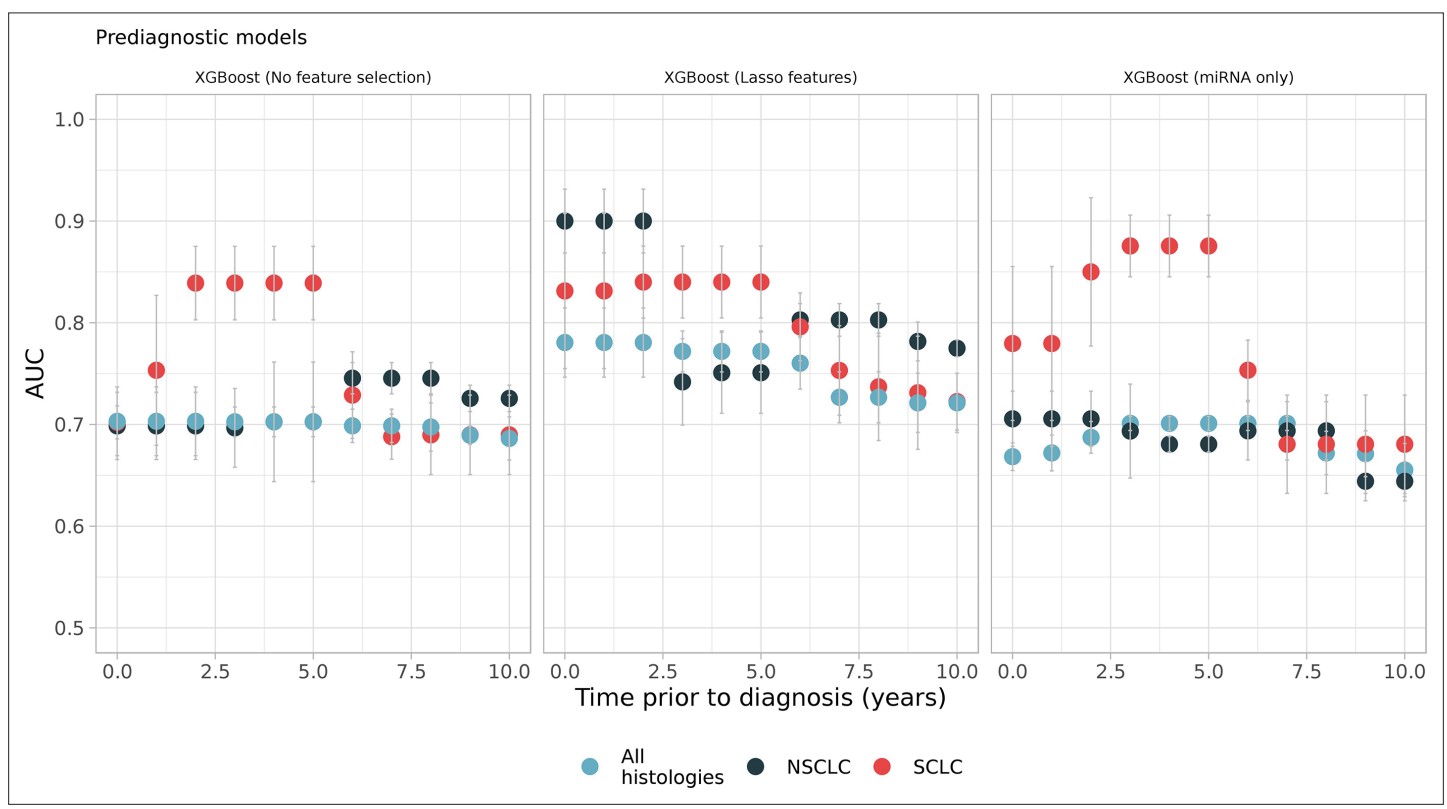

**Figure 3.** Sliding windows analysis showed better models which utilizes prediagnostic samples in specific time intervals such as small-cell lung cancer (SCLC) models, which were restricted to samples from 2 to 5 years prior to diagnosis (see the first and the second panel, red dots). Each color represents different histologies: black and red only have non-small cell lung cancer (NSCLC) and SCLC samples respectively while blue has all histologies including others (*Figure 3—source data 1*).

The online version of this article includes the following source data for figure 3:

**Source data 1.** Source data of all the panels for *Figure 3*.

## Histology-specific prediagnostic models can improve prediction performance

We previously demonstrated that RNA levels are dynamic and histology-specific in prediagnostic samples (*Umu et al., 2020*). We therefore trained and tested models stratified by prediagnostic time which were selected by a sliding window approach as explained in Materials and methods.

The results showed that inclusion of prediagnostic time and histological subtype together creates better models for specific time intervals (*Figure 3*). For example, SCLC models restricted to samples from 2 to 5 years prior to diagnosis had an average AUC of 0.84 (95% CI, 0.77–0.9) (*Figure 3*). Another model of SCLC samples that only utilized miRNAs restricted to 3–5 years prior to diagnosis had an average AUC of 0.85 (95% CI, 0.76–0.93) on the test datasets. Both SCLC models selected the same miRNAs as their most important features such as hsa-miR-30a-5p, hsa-miR-339–3p, hsa-miR-215–5p. Besides miRNAs, an isomiR of hsa-miR-451a and RN7SL181P were the most important features of prediagnostic SCLC models. Enrichment analysis of the most important features identified signaling pathways, such as MAPK, PI3K-Akt, RAS, and other pathways like choline metabolism, cellular senescence, and PD-L1 expression and PD-1 checkpoint. Similarly, NSCLC models restricted to 6–8 years prior to diagnosis had an average AUC of 0.81 (95% CI, 0.75–0.86). The most important RNAs of this period were tRF-YP9L0N4V3, an isomiR of hsa-miR-484 (iso-23-8K4P8R8SDE) and tRF-9MV47P596V. More than 70 pathways were enriched such as endocytosis, MAPK, RAS, choline metabolism, and neurotrophin signaling pathway.

**Table 3.** All selected features, performance, and relative risk (RR) of XGBoost models.

| | **Models** | | |
| --- | --- | --- | --- |
| | All* | NSCLC | SCLC |
| Features | iso-20-5KP25HFF GBP3 hsa-miR-30a-5p INTS10 LINC01362 piR-hsa-28723 RNU1-8P iso-23-BQ8DQWM4Z CTD-3252C9.4 DST HBA2 HIST2H2AC hsa-miR-99b-3p LATS1 piR-hsa-28391 piR-hsa-28394 RN7SL181P RN7SL8P RNU2-27P iso-23-8YUYFYKSY TLN1 tRF-V47P59D9 tRF-86V8WPMN1EJ3 tRF-6SXMSL73VL4Y tRF-QKF1R3WE8RO8IS | LINC01362 Y-RNA iso-23-B0NKZ01J0D iso-22-MKJIJLJ2Q iso-21-N2NBQRZ00 GBP3 iso-20-RNUW92OI GNAS hsa-miR-30a-3p NHSL2 piR-hsa-28488 RC3H2 RN7SL181P RNU2-19P RNY4P27 iso-23–909 U247N04 tRF-I89NJ4S2 tRF-9MV47P596VE tRF-86J8WPMN1EJ3 tRF-86V8WPMN1EJ3 tRF-Q1Q89P9L8422E | AC113404.1 C6orf223 HIST1H4E hsa-miR-30a-5p hsa-miR-574–5p ODC1 PTCH2 PTMA RN7SL181P tRF-22-947673FE5 AKAP9 MIGA1 RAP1B RN7SL724P RUFY2 iso-23-X3749W540L tRF-BS68BFD2 tRF-R29P4P9L5HJVE tRF-ZRS3S3$R \times$ 8HYVD |
| Total features | 25 | 21 | 19 |
| Total test samples (total leave-out size) (non-smokers) | 640 (535) (263) | 465 (360) (262) | 444 (395) (256) |
| AUC on test (95% CI) (only smokers**) | 0.76 (0.68–0.83) | 0.78 (0.70–0.85) | 0.88 (0.83–0.94) |
| AUC on test (95% CI) (both smokers and non-smokers**) | 0.68 (0.63–0.72) | 0.68 (0.63–0.73) | 0.84 (0.79–0.9) |
| RR on test (95% CI) (only smokers**) | 2.37 (1.54–3.7) $p = 1.15 \times 10^{-7}$ | 2.36 (1.52–3.66) $p = 2.83 \times 10^{-6}$ | 2.48 (2.06–3) $p = 3.32 \times 10^{-9}$ |
| RR on test (95% CI) (both smokers and non-smokers**) | 1.84 (1.7–2.01) $p = 1.25 \times 10^{-6}$ | 1.52 (1.27–1.83) $p = 2.67 \times 10^{-5}$ | 2.04 (1.85–2.25) $p = 8.8 \times 10^{-8}$ |

*Including other histologies. ** includes samples previously not used (leave-out samples).

As an alternative to sliding windows, we also performed a fixed window approach and trained models using samples from up to 2 years (0–2), up to 5 years (0–5), and up to 8 years (0–8) before diagnosis. The results showed slight improvement in model performance compared to full-time models (see *Supplementary file 3*). However, sliding windows models performed better on specific time intervals.

### Frequent features can create simple and accurate models

We created models by compiling the best features from the full-time models. Our results showed improved prediction performance for these models despite inclusion of leave-out datasets into the test set (see Materials and methods). In the test datasets including only smokers, AUC for all histologies was 0.76 (95% CI, 0.68–0.83); NSCLC model was 0.78 (95% CI, 0.70–0.85); SCLC model was 0.88 (95% CI, 0.83–0.94) (*Table 3*). However, when non-smokers were also included in the test set, the model performance dropped to 0.68 (0.63–0.72) for all histologies and 0.68 (0.63–0.73) for NSCLC. Remarkably, the SCLC model still had AUC of 0.84 (0.79–0.9) when including non-smokers.

The RRs and their associated p-values on the test dataset, with and without non-smokers, are reported in *Table 3*. A positive test in smokers suggests more than two times higher risk of getting LC diagnosis in future.

We also investigated the prediagnostic models, using the ML workflow, and selected two pairs of models for NSCLC and SCLC, which showed high performance before and after 5 years prior to diagnosis (see *Supplementary file 4*). We found that NSCLC models restricted to 0–2 and 6–8 years before diagnosis had an average AUCs of 0.89 (95% CI, 0.84–0.96) and 0.82 (95% CI, 0.76–0.88), respectively; SCLC models restricted to 2–5 and 8–10 years before diagnosis had an average AUCs of 0.89 (95% CI, 0.77–1.0) and 0.83 (95%, 0.69–0.97), respectively. We reported other model metrics and the best features in the supplementary document (*Supplementary file 4*).

## Discussion

In this study, we showed that ML models of prediagnostic serum RNA levels can be used to predict LC years before diagnosis and manifestation of disease symptoms. Our models achieved clinically relevant performance in terms of AUC, accuracy, sensitivity, and specificity (*Tables 2 and 3*). The model performance was further increased for specific prediagnostic time windows and histologies making it feasible to develop them as biomarkers for LC screening (*Figure 3*). A collection of the

| Lung cancer development | No detectable disease | Preneoplasia/Early stage tumor |
|---|---|---|
| **Proposed use of RNA biomarkers** | **Risk assessment** Biomarker monitoring of individuals (smokers) with full-time models. Positive results from any models suggest elevated risk *(at least 2 times)* while others least concern. | **Early detection** Biomarker monitoring of elevated risk individuals with prediagnostic models. Positive results suggest selection for CT monitoring and a signal between preneoplasia and early stage tumor. Histology specific models can further improve diagnosis accuracy *(mean accuracy more than 80%)*. Biyearly monitoring is recommended for others with prediagnostic models. |

**Figure 4.** Suggested clinical uses of RNA biomarkers in lung cancer (LC) screening. A positive test from full-time models shows elevated risk (at least two times). They can detect cancer-related RNA signals up to 10 years before diagnosis. Prediagnostic models have higher accuracy, sensitivity, and specificity which can potentially assist full-time models and improve specificity (*Supplementary file 4*).

The online version of this article includes the following source data for figure 4:

**Source data 1.** Suggested clinical uses of RNA biomarkers in lung cancer (LC) screening.

best models (and predictors) (*Table 3* and *Supplementary file 4*) can predict risk for developing LC, which histologies to look for and indicate the level of cancer progression. The time window of the high-performance models may be a first indication of how often to screen for LC (*Figure 4*). Our study is unique in including serum samples collected up to 10 years prior to LC diagnosis and a large set of control samples.

We previously reported that prediagnostic circulating RNA signals are highly dynamic in LC patients and they can be histology and stage dependent (*Umu et al., 2020*). In the present study, ML models using all samples regardless of stage, histology, or prediagnostic time successfully separated LC patients from controls. All the tested algorithms consistently produced acceptable AUC values (*Figure 2A*). The best algorithm, XGBoost, resulted in an average accuracy of 69% without feature selection. An analysis of the features showed a large panel of selected RNAs: more than 300 out of available 3306 (with no feature selection implemented). This may be interpreted as a general shift in the levels of RNAs during cancer development, consistent with our previous study that showed hundreds of RNAs were differentially expressed up to 10 years before diagnosis (*Umu et al., 2020*).

We found that some features were considerably more important (and frequent) separators than others with or without feature selection. The list includes piRNAs (e.g. piR-hsa-28723), miRNAs (e.g. hsa-miR-574–5p, hsa-miR-30a-5p, hsa-miR-106b-5p), isomiRs (e.g. isomiR of hsa-miR-423–5p (iso-20-5KP25HFF), hsa-miR-486–5p (iso-23-8YUYFYKSY)), and miscRNAs (e.g. RN7SL181P). Some of them were particularly interesting since they were associated with cancer or proposed as cancer biomarkers. Hsa-miR-30a-5p is a tumor suppressor and downregulated in LC tissues (*Yanaihara et al., 2006*). It regulates oncogenes such as *RAB38* and *RAB27B* (*The RNAcentral Consortium, 2019*). Another notable example is hsa-miR-574–5p which promotes metastasis in NSCLC by targeting *PCP2* in tumor tissues (*Zhou et al., 2016*) and has been proposed as an early stage NSCLC serum biomarker (*Foss et al., 2011*). Hsa-miR-574–5p was among the most important features in lasso-selected and miRNA-only histology-specific SCLC models. It was also one of the most important features in our histology-specific NSCLC models. There were also isomiRs among the most important features such as hsa-miR-486–5p canonical form, which was the best separator for all histologies. Hsa-miR-486–5p targets *PIK3R1* to suppress cell growth. Its overexpression inhibits cell proliferation and invasion and it was significantly downregulated in both tissue and serum (*Tian et al., 2019*). Hsa-miR-486–5p was proposed as a diagnostic and prognostic biomarker for NSCLC (*ElKhouly et al., 2020*; *Tian et al., 2019*).

Besides miRNAs and isomiRs, RNAs of other classes were noteworthy and linked to carcinogenesis. For example, 7SL, a member of miscRNAs, is upregulated in tumor cells. It binds to *TP53* mRNA at the 3'UTR region and downregulates its expression (*Abdelmohsen et al., 2014*). 7SL-related transcripts (e.g. RN7SL181P) were among the most important separators in the cell histology, NSCLC- and SCLC-specific models. Another example is Y-RNA and we found that Y-RNA and related genes (e.g. RNY4P30) were among the most important features for NSCLC models. Y-RNA was also chosen as an important feature by the lasso-selected NSCLC models. Y-RNA-derived small RNAs function as tumor suppressors in NSCLC. They inhibit cell proliferation and were proposed as circulating RNA biomarkers since they were upregulated in NSCLC EVs (*Li et al., 2018*).

Inclusion of both prediagnostic time and histology produced better models in certain time windows (e.g. 2–5 years before diagnosis for SCLC) (*Figure 3*). This can be explained by the dynamic nature of prediagnostic RNA levels (*Lund et al., 2016*; *Umu et al., 2020*). Important features of these models can also be linked to early carcinogenesis and some were specific to these models. For example, hsa-miR-339–3p was among the most important features of SCLC prediagnostic models. Hsa-miR-339–3p is a tumor suppressor and was proposed as a serum biomarker of LC (*Yu et al., 2019*). We retrained some of these prediagnostic models using the most frequent features and achieved higher prediction performance than the full-time models in specific time intervals. We reported these models in supplementary (*Supplementary file 3*).

The most important features of histology-specific models also showed associations with carcinogenesis-related KEGG pathways, which were common or specific to histology. The common ones include well-known signaling (e.g. RAS, PI3K-Akt, MAPK, ErbB) and cancer-related pathways (e.g. proteoglycans in cancer and pathways in cancer). Choline metabolism in cancer pathway was one of the common ones and enriched in some histology-specific prediagnostic models. Altered choline profiles are characteristics of tumor tissues (*Glunde et al., 2006*). Moreover, a lipidome serum

profiling study on early stage NSCLC patients proposed choline-containing phospholipids as potential LC biomarkers (*Klupczynska et al., 2019*). Enrichment of choline metabolism pathway years before diagnosis (i.e. NSCLC 6–8 and SCLC 2–5) supports this conclusion. We also reported enrichment of this pathway for all histologies before diagnosis in our previous study (*Umu et al., 2020*).

A strength of our study is the large sample size from prediagnostic cases and a large control group from cancer-free individuals from the same cohort. We have detailed information on histological subtype and stage at diagnosis from the Cancer Registry of Norway (CRN) and smoking history from survey data. We also accounted for other potential confounders (i.e. age, sex, and blood donor group [BDg]) (*Rounge et al., 2018*). Some of our potential biomarkers (e.g. hsa-miR-30a-5p, sa-miR-339–3p, 7SL) were already associated with carcinogenesis or proposed as biomarkers, which shows consistent results with current literature. Further, we found potential biomarkers from overlooked RNA classes which add important new knowledge into the field. We shared the average feature importance of all RNAs as supplementary tables (*Supplementary file 2*). We investigated performance of different algorithms which showed consistent results in terms of AUCs and features. We compiled shortlists from the most important features and tested their performance in a leave-out dataset on both smokers and non-smokers. We also found that smokers with a positive test had more than two times higher risk of getting LC diagnosis in future (*Table 3*).

There are some weaknesses in our study that we need to address. First, an independent cohort should replicate our results. However, only a few cohorts include prediagnostic samples that can be used for discovery and validation. We tried to overcome this issue by using training-testing repeats for assessing generalisability. We also reported our results with and without feature selection since some feature selection methods (e.g. lasso and univariate) can cause overfitting. Second, using more than one sample from the same individual can potentially cause overfitting. However, we did not detect any effect related to this issue (*Figure 2—figure supplement 2*). Third, our study focused only on smokers (since case samples are mostly smokers). However, our results show acceptable performance when including non-smokers as a test dataset as well. Fourth, reuse of the same data for frequent biomarker models (as reported in *Table 3*) can also result in overfitting. We tried to overcome this issue by including a leave-out dataset (which was never used) into the test set and reported performance. Lastly, since our samples are long-term stored, some unstable RNA molecules may have been degraded over the years, though we have already shown that this effect is negligible (*Umu et al., 2018*). Yet, we matched cases and controls for BDg which includes the effect of storage time (see Materials and methods).

In LC screening programs, RNA biomarkers can be used as a tool of initial assessment or combined with LDCT for early detection (*Hanash et al., 2018*). We found that smokers with a positive test had higher risk of getting LC diagnosis in future (*Table 3*). We also found that our biomarkers can be potentially used on non-smokers, especially SCLC biomarkers. However, we do not have enough non-smoker cases to further validate this interpretation. The dynamic nature of the prediagnostic signal for cancer may pose challenges for the performance of modeling and biomarker development. However, using a set of models specific for histology and time might provide additional information useful in evaluating LC risk (*Figure 4*). Our proposed use of RNA biomarkers starts with risk assessment using standard full-time models which can be used for an initial assessment in smokers when the disease is undetectable. A positive signal (i.e. high probability of being in LC group) classifies those individuals into an elevated risk group. Since prediagnostic models have a 2-year peak performance, every second-year testing with these models can provide confirmation of preneoplasia or an early stage tumor for individuals with elevated risk and selection criteria for CT monitoring. Prediagnostic models had higher overall specificity (more than 80%) which can help to determine future diagnosis histology. However, it requires further research. We selected two sets of histology-specific diagnostic models for early/late NSCLC and SCLC diagnosis and reported these in the supplementary document (*Supplementary file 4*). RNA biomarkers can prevent unnecessary use of LDCT while improving the chance of an early diagnosis of LC in an early stage. This hypothesis can be investigated in screening programs for validation.

## Conclusion

We have shown that LC can be detected in both smokers and non-smokers years before diagnosis and the manifestation of symptoms regardless of histological subtype. We also proposed a model on how

RNA biomarkers can be utilized in clinical settings. Our top performing models can produce AUCs up to 0.9 before diagnosis suggesting a great potential for LC early prediction.

## Materials and methods

### Study population and data sources

We used the population-based Janus Serum Bank (JSB) cohort containing prediagnostic serum samples (*Langseth et al., 2017*). The study participants were identified by linking the JSB to the CRN. We restricted our analyses to patients later diagnosed with LC up to 10 years after blood donation and control samples from individuals cancer-free (except non-melanoma skin cancer) at least 10 years after sample collection. We matched cases and controls on confounders (see Bioinformatics analyses). Smoking, collected from health survey data, was classified as current, former, or never smokers (*Hjerkind et al., 2017*). Since we have previously shown that smoking significantly affects RNA expression levels, we only included smokers (i.e. current and former) in the initial analyses and model building. However, non-smokers and samples not included by the frequency matching were used as an additional independent leave-out dataset to assess the level of overfitting.

### Tumor staging

Detailed cancer information was selected from the CRN that has systematically collected mandatory notification on cancer occurrence for the Norwegian population since 1952 (*Larsen et al., 2009*). The cases were classified into histological subtypes: NSCLC, SCLC, and others, the latter referring to other less defined or multiple histologies. Stage at diagnosis was encoded with the TNM system: early (localized – stage I), locally advanced (regional – stages II and III), advanced or metastatic (distant – stage IV), and unknown (*Cancer Registry of Norway, 2020*).

### Laboratory processing

We extracted RNA from 400 µL serum using phenol-chloroform and miRNeasy Serum/Plasma kit (Qiagen, Valencia, CA). We performed size selection using a 3% Agarose Gel Cassette (Cat. No CSD3010) on a Pippin Prep (Sage Science) with a cut size optimized to cover RNA molecules from 17 to 47 nt in length. Libraries were prepared with the NEBNext Small RNA kit (NEB, Ipswich, MA) and sequenced on a HiSeq 2500 platform to on average 18 million sequences per sample (Illumina, San Diego, CA).

### Bioinformatic analyses

Our bioinformatics workflow includes quality control, adapter trimming, read mapping, read counting, and creation of count tables. We used a large annotation dataset containing several RNA classes available in serum (*Umu et al., 2018*), including miRBase (v22.1) for miRNAs (*Kozomara et al., 2019*), piRBase (v1.0) for piRNAs (*Zhang et al., 2014*), and the GENCODE (v26) for other RNA classes (*Harrow et al., 2012*). We used the AdapterRemoval tool for adapter trimming (*Schubert et al., 2016*) and Bowtie2 (*Langmead and Salzberg, 2012*) for mapping reads to the human genome (hg38) with an average mapping ratio of 70%. The SeqBuster tool was used for miRNA annotation counts and isomiR calling (*Pantano et al., 2010*). We filtered out the RNAs with fewer than five reads in less than 80% of the samples. All isomiRs passed the expression were regarded as bona fide isomiRs. We used DESeq2's (*Love et al., 2014*) variance stabilizing normalization function to normalize identified RNA counts. The *optmatch* (v0.9–11) R package (*Hansen and Klopfer, 2006*) selected appropriately matched controls while building models. Therefore, we matched LC samples and controls on sex, age at donation, and BDg. BDg is a technical cofounder combining the effect of sample treatment at donation and storage time (*Rounge et al., 2018*). We used R function kegga from the limma package for KEGG pathway enrichment analysis of selected RNA features if they are miRNA, isomiR, or mRNA. The miRNA and isomiR targets were extracted from MIRDB (v5.0) predictions (*Wong and Wang, 2015*) (score cutoff >60). p-Values were adjusted using false discovery rate (FDR) (using *p.adjust* function of R).

## ML classification algorithms and training/testing workflow

High dimensionality is often a problem in modeling RNA-seq data. Our preliminary analysis showed that ML algorithms with regularization produced successful models. Therefore, we selected five ML algorithms to create our initial models: lasso, elastic-net, sparse group lasso (SGL), RF, and extreme gradient boosting (XGBoost) algorithms. We used fivefold cross-validation (if available) to tune hyperparameters for model training. For the SGL models, RNAs were classified by type.

R implementations of these algorithms were used: *caret* (v6.0–84) and *glmnet* (v2.0–18) packages for elastic-net and the lasso, *sglfast* (v0.10) and *msgl* (v2.3.9) for the SGL models and *xgboost* (v1.0.0.2) for XGBoost. Classifications were performed according to histology and time to diagnosis (for details see next paragraph) using an automated ML workflow. In the ML workflow the datasets were split into training (70%) and test (30%) (*Figure 1B*). We repeated this step five times using designated seed numbers to select five different training and test datasets which were balanced for case/control numbers and also matched for confounders (i.e. sex, age, and BDg). Model optimization including hyperparameter tuning was done by a grid search approach followed by fivefold cross-validation using the training sets. The test datasets were only used for testing to overcome overfitting and assess true performance. The performance of the classifiers were mainly evaluated by area under the ROC curves (AUC)s. We also calculated accuracy, sensitivity, and specificity. Confidence interval calculations were done using metrics of test datasets.

## Histology and prediagnostic models

We refer to models for all histologies, NSCLC and SCLC that do not take time to diagnose into account as standard full-time models (*Figure 1B*). Prediagnostic models were created using a sliding windows approach and a fixed-time approach to find optimal time to diagnose intervals. We first selected three different window sizes, 2, 3, and 4 years, which were moved over the 10 years prior to diagnosis time. We then built models based on samples captured by these sliding windows. Fixed-time windows were 0–2, 0–5, and 0–8 years before diagnosis. We used the workflow described above to train and test both standard and prediagnostic models.

## Feature selection methods

We implemented feature selection methods to improve model performances, including single-RNA class, lasso selection, and significant selection. In the single-RNA class method, we dropped all RNA types except one. In lasso selection, all non-zero features selected by the lasso classification models were pooled. Next, we retrained new classification models which were restricted to use only these features. In significance selection, an univariate regression analysis was done per feature and significant features (multiple testing adjusted) were used to train classification models.

## Frequent feature models, independent leave-out test, and RR calculations

We created models for each histology which utilize the most frequent features identified in the standard full-time models. To assess overfitting and to get a better estimate of these model performances, we split the datasets into training (80%) and test (20%) sets. To the test sets we also added non-smokers and samples from smokers, but not previously used in frequency matching (number of samples reported in *Table 3*). Both unmatchable samples and non-smokers were never used for model building and evaluation which we refer to as leave-out sets. We did not repeat this analysis five times as in the automated ML workflow. RRs were calculated using the test sets. The optimal threshold was identified in cross-validation. We used the R packages *cutpointr* (v1.0.1) and *epitools* (v0.5–10.1) to calculate RRs.

## Data accessibility

The datasets generated for this article are not readily available because of the principles and conditions set out in articles 6 (1) (e) and 9 (2) (j) of the General Data Protection Regulation (GDPR). National legal basis as per the Regulations on population-based health surveys and ethical approval from the Norwegian Regional Committee for Medical and Health Research Ethics (REC) is also required. Requests to access the datasets should be directed to the corresponding authors. Our scripts and

bioinformatics workflow files can be accessed from our GitHub repo (https://github.com/sinanugur/LCscripts, *Umu, 2022* copy archived at swh:1:rev:26bccc86a551f71284559db11bb74230f5d00cc4).

## Acknowledgements

This work was supported by The Norwegian Research Council's Programme 'Human Biobanks and Health Data [229621 /H10, 248791 /H10]. Disclosure of invention was accepted by the technology transfer office, Invent2 (DOFI: 19010). We would like to acknowledge Cecilie Bucher-Johannessen, Marianne Lauritzen, Magnus Leithaug for performing lab and coordination tasks. We also acknowledge Matthew D Whitaker and Marc Chadeau-Hyam from Imperial College London for discussions on ML model training and testing. We acknowledge the Norwegian Institute of Public Health for access to survey data in this study. The sequencing service was provided by the Norwegian Sequencing Centre (https://www.sequencing.uio.no), a national technology platform hosted by Oslo University Hospital and the University of Oslo supported by the Research Council of Norway and the Southeastern Regional Health Authority.

## Additional information

### Funding

| Funder | Grant reference number | Author |
|---|---|---|
| The Research Council of Norway | Human Biobanks and Health Data [229621/H10] | Hilde Langseth Trine B Rounge |
| The Research Council of Norway | Human Biobanks and Health Data [248791/H10] | Hilde Langseth Trine B Rounge |

The funders had no role in study design, data collection and interpretation, or the decision to submit the work for publication.

### Author contributions

Sinan U Umu, Formal analysis, Investigation, Methodology, Software, Writing – original draft; Hilde Langseth, Conceptualization, Funding acquisition, Project administration, Writing - review and editing; Verena Zuber, Methodology, Writing - review and editing; Åslaug Helland, Writing - review and editing; Robert Lyle, Resources, Writing - review and editing; Trine B Rounge, Conceptualization, Funding acquisition, Project administration, Writing – original draft, Writing - review and editing

### Author ORCIDs

Sinan U Umu (ID) http://orcid.org/0000-0001-8081-7819
Trine B Rounge (ID) http://orcid.org/0000-0003-2677-2722

### Ethics

Human subjects: This study was approved by the Norwegian Regional Committee for medical and health research ethics (REC no: 19892 previous 2016/1290) and was based on broad consent from participants in the Janus cohort. The work has been carried out in compliance with the standards set by the Declaration of Helsinki.

### Decision letter and Author response

Decision letter https://doi.org/10.7554/eLife.71035.sa1
Author response https://doi.org/10.7554/eLife.71035.sa2

## Additional files

### Supplementary files

• Supplementary file 1. Clinical and histological characteristics of non-smoker samples of leave-out dataset.
• Supplementary file 2. Detailed feature importance tables for all trained models.

- Supplementary file 3. Fixed-time model performance on different histologies.
- Supplementary file 4. Selected prediagnostic models, metrics, and their feature importance tables.
- Transparent reporting form

### Data availability

The datasets generated for this manuscript are not readily available because of the principles and conditions set out in articles 6 (1) (e) and 9 (2) (j) of the General Data Protection Regulation (GDPR). National legal basis as per the Regulations on population-based health surveys and ethical approval from the Norwegian Regional Committee for Medical and Health Research Ethics (REC) is also required. Requests to access the datasets should be directed to the corresponding authors with a project proposal. Please refer to our project website for the latest information on data sharing (kreftregisteret. no/en/janusrna). Our scripts, plot data, and bioinformatics workflow files can be accessed from our Github repo (https://github.com/sinanugur/LCscripts copy archived at swh:1:rev:26bccc86a551f7128 4559db11bb74230f5d00cc4).

The following dataset was generated:

| Author(s) | Year | Dataset title | Dataset URL | Database and Identifier |
|---|---|---|---|---|
| Umu SU | 2021 | Lung Cancer analyses scripts | https://github.com/ sinanugur/LCscripts | GitHub, 439cf34 |

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
