## [Editor Report]

This work has generated valuable data demonstrating the potential utility of serum RNA for lung cancer detection.

---

## [Decision Letter]

**Decision letter after peer review:**

Thank you for submitting your article "Serum RNAs can predict lung cancer up to 10 years prior to diagnosis" for consideration by *eLife*. Your article has been reviewed by 2 peer reviewers, and the evaluation has been overseen by Y M Dennis Lo as the Senior Editor/Reviewing Editor. The following individual involved in review of your submission has agreed to reveal their identity: Shenglin Huang (Reviewer #2).

The reviewers have seen each other's reviews, and the Reviewing Editor has drafted this to help you prepare a revised submission.

Essential revisions:

1. The analysis and result of serum RNA sequencing were not clearly presented. RNA sequencing with an average of 18 million reads per sample was performed. How about the mapped rate? Since the samples are long-term stored and may be at different times of collection, it is important that the sequencing results should be consisent. How about the sample correlations? In addition, why many of the RNA candidates (1137 of 3306) are isomiRs? They are derived from miRs, or degradation?

2. The authors showed that the candidate biomarkers were invovled in the cancer-related pathways. This result is somewhat overestimated because these RNAs are small RNAs and the KEGG pathway enrichment analysis is based on the targets.

3. A strength of this study is the large sample size from prediagnostic cases and a large control group from cancer-free individuals from the same cohort. The authors obtained RNA-seq data from 1061 serum samples. However, only samples from a total of 645 individuals with smoking history were analyzed. It would be helpful to include the samples without smoking to investigate the specificity of the potential biomarkers.

4. There are concerns regarding multiple samples from the same cancer subject. If samples from one subject go to both the training and testing set, there would be issues of overfitting. In the revision, the authors need to remove the duplicate samples from the analysis.

5. Calculate the relative risk for future cancer development for individuals with positive and negative test results.

6. Instead of using sliding windows of fixed duration, use windows of increasing duration.

7. Address the issue of over-fitting for the analysis from line 338 onwards.

8. The proposed use of the analysis (Figure 5 and the discussion) does not appear to be realistic. The authors need to calculate the percentage of subjects falling into the test positive and negative groups based on their analysis. Then, for each group, what percentage of subjects will eventually develop LC.

*Reviewer #2:*

This study by Umu et al., investigated the biomarker potential of serum RNAs for the early detection of lung cancer (LC) in smokers at different prediagnostic time intervals and histological subtypes. They observed that smokers later diagnosed with LC can be robustly separated from healthy controls by using machine learning models with serum RNAs profiles. This work demonstrated that serum RNAs could be promising prediagnostic biomarkers in a LC screening setting. This finding is interesting but not convincing because only one sample cohort was evaluated in current study and the datasets generated for this manuscript are not available. Also there are some concerns as follows.

1.The analysis and result of serum RNA sequencing were not clearly presented. RNA sequencing with an average of 18 million reads per sample was performed. How about the mapped rate? Since the samples are long-term stored and may be at different times of collection, it is important that the sequencing results should be consisent. How about the sample correlations? In addition, why many of the RNA candidates (1137 of 3306) are isomiRs? They are derived from miRs, or degradation?

2.The authors showed that the candidate biomarkers were invovled in the cancer-related pathways. This result is somewhat overestimated because these RNAs are small RNAs and the KEGG pathway enrichment analysis is based on the targets.

3.A strength of this study is the large sample size from prediagnostic cases and a large control group from cancer-free individuals from the same cohort. The authors obtained RNA-seq data from 1061 serum samples. However, only samples from a total of 645 individuals with smoking history were analyzed. It would be helpful to include the samples without smoking to investigate the specificity of the potential biomarkers.

*Reviewer #3:*

The authors analyzed archived serum samples which were collected before the development of lung cancer (LC) for different species of RNA.

They used machine learning to investigate if the combination of the RNA markers would be useful for the prediction of the future development of LC. In the machine learning analysis, they used 70% samples for training and 30% samples for testing. The process was repeated 5 times so that the samples can be assigned to both training and testing groups.

The strength of the study is that they include a large cohort of samples collected years before the development of LC.

The weaknesses of the study include

1. the questionable biological basis of this approach. As the samples were collected years before cancer diagnosis, the amount of tumor-derived RNA is expected to be very low. In this regard, the aberrations in serum RNA, if any, are likely to reflect the background risk of the body as a whole.

2. multiple samples from the same cancer subject. If samples from one subject go to both the training and testing set, there would be issues of overfitting. The authors would need to remove the duplicate samples from the analysis.

3. the low diagnostic performance. The sensitivity and specificity of the models were both around 70%. This would not be useful clinically. In fact, the use of relative risk for individuals with positive and negative test results would be more useful for predictive models.

4. The authors used a sliding window of fixed duration for the analysis. This analysis is counter-intuitive. If a marker is useful for predicting cancer at 3-5 years before cancer, what is the reason why it cannot predict cancers that occur within 3 years. It would be better to use windows with flexible size e.g. prediction of cancer in the next 2, then next 3, then next 5 years, etc.

5. For the analysis on lines 338 onwards, the authors further select features identified in the previous machine learning models for evaluation. What is the testing group in this analysis? Did they use all the samples without separating training and testing groups? If so, the over-fitting problem would be significant.

6. The. proposed use of the analysis (Figure 5 and the discussion) does not appear to be realistic. The authors need to calculate the percentage of subjects falling into the test positive and negative groups based on their analysis. Then, for each group, what percentage of subjects will eventually develop LC.

7. Remove duplicated samples from the LC group.

8. Calculate the relative risk for future cancer development for individuals with positive and negative test results.

9. Instead of using sliding windows of fixed duration, use windows of increasing duration.

10. Address the issue of over-fitting for the analysis from line 338 onwards.

11. To address the issue of clinical utility raised in point 6 in the previous section.

---

## [Author Response]

Essential revisions:1. The analysis and result of serum RNA sequencing were not clearly presented. RNA sequencing with an average of 18 million reads per sample was performed. How about the mapped rate? Since the samples are long-term stored and may be at different times of collection, it is important that the sequencing results should be consisent. How about the sample correlations? In addition, why many of the RNA candidates (1137 of 3306) are isomiRs? They are derived from miRs, or degradation?

We have now improved the presentation of the analysis and results throughout the manuscript. The manuscript builds on previous publications showing feasibility and profiles of serum small RNA sequencing, confounding, long-term stability and differential expression in cancer in this biobank (Rounge et al., CEBP 2015, Umu et al., RNA Biology, 2018, Rounge et al., Sci Rep, 2018, Umu et al., Mol Oncol, 2020 and Burton et al., Front in Oncol, 2020).

However, we agree with the reviewer that more information about the characteristics of the RNA profiles would be helpful to the reader. To answer the specific questions: First, the average mapping rate was around 70%. We revised the materials methods section “*laboratory processing and bioinformatic analyses*” and included the information on mapping tools and rates. We created a separate section called “Bioinformatic analyses” (see page 4 and 5).

Regarding long-term storage, we already showed that long-term stored samples are a viable source for small RNA analysis (Rounge et al., 2015). Small RNAs (e.g. miRNA) are stable during long-term archiving (Umu et al., 2018). We showed that some traits such as sex, age of patients and blood donor group (BDg) affect RNA profiles. Therefore, these traits must be matched between cases and controls. BDg refers to a technical cofounder combining the effect of sample treatment at donation and storage time (Rounge et al., 2018). In this revision, we further explained this and added these sentences into Materials and methods:

“Therefore, we frequency matched LC samples and controls on sex, age at donation, and blood donor group (BDg). BDg is a technical cofounder combining the effect of sample treatment at donation and storage time (Rounge et al., 2018).” (see page 4)

In Umu et al., 2018, we also investigated whether RNAs were degradation products or not. We found that RNA fragments, such as tRNA derived fragments, had specific expression patterns, which can be explained by RNA biogenesis. We concluded that many RNA fragments in serum must be biological products, including tRFs and isomiRs. Therefore, in our submitted manuscript, we used the SeqCluster (miraligner) tool and we regarded all isomiRs which passed the expression threshold as bona fide isomiRs. In this version, we also included mirTOP identifiers of candidate isomiRs.

Our definition of expression threshold is:

“any RNAs with more than 5 reads mapped in 80% of the samples”.

This is a high threshold to detect robust signals of expression. We now cited SeqCluster and mirTOP and included our expression detection threshold in Materials and methods, which was missing in our initial submission. (see page 4)

2. The authors showed that the candidate biomarkers were involved in the cancer-related pathways. This result is somewhat overestimated because these RNAs are small RNAs and the KEGG pathway enrichment analysis is based on the targets.

Thank you for pointing this out. miRNAs can target multiple genes. Unfortunately there is no easy way to do pathway enrichment analysis for other important small RNAs (e.g. miscRNAs), which are underrepresented. We believe it is a plausible assumption to expect miRNAs will disrupt a large set of biological pathways including cancer related ones since they often regulate multiple mRNAs.

While predicting miRNA targets, we used a high threshold to reduce the total number of false positives. We also performed a multiple testing correction to filter out less significant enrichments. Our literature search (as discussed in the results and discussion) also showed that many biomarker candidates, such as miRNAs and some other sncRNAs, were associated with cancer-related pathways.

3. A strength of this study is the large sample size from prediagnostic cases and a large control group from cancer-free individuals from the same cohort. The authors obtained RNA-seq data from 1061 serum samples. However, only samples from a total of 645 individuals with smoking history were analyzed. It would be helpful to include the samples without smoking to investigate the specificity of the potential biomarkers.

As reviewers suggested, we included non-smokers for AUC estimation and relative risk calculation. This also assessed the robustness and overfitting issue that was brought by the reviewers (please refer to question 7). Non-smokers were not used for training and only used to evaluate frequent biomarkers. We put non-smokers characteristics table into the supplementary document (Table S1).

4. There are concerns regarding multiple samples from the same cancer subject. If samples from one subject go to both the training and testing set, there would be issues of overfitting. In the revision, the authors need to remove the duplicate samples from the analysis.

To answer this question, we repeated the XGBoost modelling described in the Results section “ML algorithms can differentiate between prediagnostic cases and controls regardless of prediagnostic time” without multiple samples. We selected only one sample per patient. We used the identical modelling workflow. Our results showed similar performance in terms of AUCs. We therefore concluded that multiple samples do not cause overfitting or any bias (see supplementary document, Figure S2).

We discussed this result briefly in the main manuscript by writing:

“Second, using more than one sample from the same individual can potentially cause overfitting. However, we did not detect any effect related to this issue (Figure S2).” (see page 18)

5. Calculate the relative risk for future cancer development for individuals with positive and negative test results.

We addressed this issue and calculated the relative risk (Table 3). We explained how we calculate the relative risk in Materials and methods. (Please also see revision answer 7)

6. Instead of using sliding windows of fixed duration, use windows of increasing duration.

To address this question, we used a fixed duration approach as suggested by the reviewers. We used three different time windows 0 to 2 years, 0 to 5 years and 0 to 8 years before diagnosis. These points were selected because the results showed slight improvement in the model performance in some time windows. The performance is not as good as sliding window models.

However, we included the results from fixed-duration models to the supplementary document as a separate section (Table S3) and briefly discussed the results in main manuscript by adding this paragraph:

“As an alternative to sliding windows, we also performed a fixed window approach and trained models using samples from up to 2 years (0-2), up to 5 years (0-5) and up to 8 years (0-8) before diagnosis. The results showed slight improvement in model performance compared to full time models (see supplementary document and Table S1). However, sliding windows models performed better on specific time intervals.”.

7. Address the issue of over-fitting for the analysis from line 338 onwards.

We agree with the reviewer that overfitting might be an issue. We tried to overcome this by including an independent dataset that was never used for testing or training.

Our machine learning workflow balances for case/control numbers and also matches samples in training and test based on “age”, “sex” and “BDg”. (Please see revision answer 1 and page 4 of the manuscript). Therefore, we previously discarded unmatchable samples for each strata and these samples were never not used. We have now utilized them to assess model performance and to assess overfitting by adding them into the testing group for section “Frequent features can create simple and accurate models”.

Furthermore, we added non-smokers into the testing dataset. We measured evaluation metrics on test datasets which returned a better estimate on the model performance.

In short, the training group contains 80% and the test group contains 20% plus smokers that have never been used for training (i.e. unmatchable samples) and non-smokers that we did not use in our original submission. We refer to these samples as independent leave-out dataset. We showed in Table 3 how many of the test set samples were never used. However, overfitting might still be an issue but we wanted to find and report a shortlist of the best biomarkers for future research.

8. The proposed use of the analysis (Figure 5 and the discussion) does not appear to be realistic. The authors need to calculate the percentage of subjects falling into the test positive and negative groups based on their analysis. Then, for each group, what percentage of subjects will eventually develop LC.

This is a good suggestion and to accommodate the reviewers comments, we have updated Figure 4 and the discussion. Please see the highlighted text (page 19).

Reviewer #2:This study by Umu et al., investigated the biomarker potential of serum RNAs for the early detection of lung cancer (LC) in smokers at different prediagnostic time intervals and histological subtypes. They observed that smokers later diagnosed with LC can be robustly separated from healthy controls by using machine learning models with serum RNAs profiles. This work demonstrated that serum RNAs could be promising prediagnostic biomarkers in a LC screening setting. This finding is interesting but not convincing because only one sample cohort was evaluated in current study and the datasets generated for this manuscript are not available. Also there are some concerns as follows.

Unfortunately the datasets generated for this article are not readily available because of the principles and conditions set out in articles 6 (1) (e) and 9 (2) (j) of the General Data Protection Regulation (GDPR). However, in a collaborative setting, data is available and can be shared in a secure environment.

1.The analysis and result of serum RNA sequencing were not clearly presented. RNA sequencing with an average of 18 million reads per sample was performed. How about the mapped rate? Since the samples are long-term stored and may be at different times of collection, it is important that the sequencing results should be consisent. How about the sample correlations? In addition, why many of the RNA candidates (1137 of 3306) are isomiRs? They are derived from miRs, or degradation?

Please see the essential revisions.

2.The authors showed that the candidate biomarkers were invovled in the cancer-related pathways. This result is somewhat overestimated because these RNAs are small RNAs and the KEGG pathway enrichment analysis is based on the targets.

Please see the essential revisions.

3.A strength of this study is the large sample size from prediagnostic cases and a large control group from cancer-free individuals from the same cohort. The authors obtained RNA-seq data from 1061 serum samples. However, only samples from a total of 645 individuals with smoking history were analyzed. It would be helpful to include the samples without smoking to investigate the specificity of the potential biomarkers.

Please see the essential revisions.

Reviewer #3:The authors analyzed archived serum samples which were collected before the development of lung cancer (LC) for different species of RNA.They used machine learning to investigate if the combination of the RNA markers would be useful for the prediction of the future development of LC. In the machine learning analysis, they used 70% samples for training and 30% samples for testing. The process was repeated 5 times so that the samples can be assigned to both training and testing groups.The strength of the study is that they include a large cohort of samples collected years before the development of LC.The weaknesses of the study include1. the questionable biological basis of this approach. As the samples were collected years before cancer diagnosis, the amount of tumor-derived RNA is expected to be very low. In this regard, the aberrations in serum RNA, if any, are likely to reflect the background risk of the body as a whole.

We agree with the reviewer, the origin of serum RNAs can be different tissues including immune cells or tumor cells. After diagnosis, the size of the tumor can still be too small to detect any originating RNAs. It is plausible that we detect RNAs originating from any tissues which signal cancer progression. We included a few sentences and citations that discuss the origin of circulating RNAs as well (see introduction, second paragraph).

As the reviewer suggested, we calculated the RR of a positive prediction.

2. multiple samples from the same cancer subject. If samples from one subject go to both the training and testing set, there would be issues of overfitting. The authors would need to remove the duplicate samples from the analysis.

As the reviewer suggested, we tested this effect and added the results into the Supplementary documents. Please also see the essential revisions answer 4.

3. the low diagnostic performance. The sensitivity and specificity of the models were both around 70%. This would not be useful clinically. In fact, the use of relative risk for individuals with positive and negative test results would be more useful for predictive models.

Please see the essential revisions.

4. The authors used a sliding window of fixed duration for the analysis. This analysis is counter-intuitive. If a marker is useful for predicting cancer at 3-5 years before cancer, what is the reason why it cannot predict cancers that occur within 3 years. It would be better to use windows with flexible size e.g. prediction of cancer in the next 2, then next 3, then next 5 years, etc.

Please see the essential revisions.

5. For the analysis on lines 338 onwards, the authors further select features identified in the previous machine learning models for evaluation. What is the testing group in this analysis? Did they use all the samples without separating training and testing groups? If so, the over-fitting problem would be significant.

Please see the essential revisions.

6. The. proposed use of the analysis (Figure 5 and the discussion) does not appear to be realistic. The authors need to calculate the percentage of subjects falling into the test positive and negative groups based on their analysis. Then, for each group, what percentage of subjects will eventually develop LC.

Please see the essential revisions.